# The development of brain pericytes requires expression of the transcription factor *nkx3.1* in intermediate precursors

**Suchit Ahuja**[1,2], **Cynthia Adjekukor**[1,2], **Qing Li**[1,2], **Katrinka M. Kocha**[1,2], **Nicole Rosin**[3], **Elodie Labit**[3], **Sarthak Sinha**[3], **Ankita Narang**[2], **Quan Long**[1,2], **Jeff Biernaskie**[2,3], **Peng Huang**[1,2], **Sarah J. Childs**[1,2]*

1 Department of Biochemistry and Molecular Biology, University of Calgary, Calgary, Canada, 2 Alberta Children's Hospital Research Institute, University of Calgary, Calgary, Canada, 3 Department of Comparative Biology and Experimental Medicine, Faculty of Veterinary Medicine, University of Calgary, Calgary, Canada

* schilds@ucalgary.ca

**Data Availability Statement:** All relevant data are within the paper and its Supporting information

## Abstract

Brain pericytes are one of the critical cell types that regulate endothelial barrier function and activity, thus ensuring adequate blood flow to the brain. The genetic pathways guiding undifferentiated cells into mature pericytes are not well understood. We show here that pericyte precursor populations from both neural crest and head mesoderm of zebrafish express the transcription factor *nkx3.1* develop into brain pericytes. We identify the gene signature of these precursors and show that an *nkx3.1*-, *foxf2a*-, and *cxcl12b*-expressing pericyte precursor population is present around the basilar artery prior to artery formation and pericyte recruitment. The precursors later spread throughout the brain and differentiate to express canonical pericyte markers. Cxcl12b-Cxcr4 signaling is required for pericyte attachment and differentiation. Further, both *nkx3.1* and *cxcl12*b are necessary and sufficient in regulating pericyte number as loss inhibits and gain increases pericyte number. Through genetic experiments, we have defined a precursor population for brain pericytes and identified genes critical for their differentiation.

## Introduction

Endothelial cells and pericytes are key partners in the brain microvessel network. Endothelial cells line the luminal side of vessels, and pericytes attach to the abluminal side of endothelial cells. Pericytes stabilize microvessels by laying extracellular matrix around endothelial cells and regulating vascular tone [1–3]. In addition, pericytes regulate postnatal endothelial sprouting and endothelial morphogenesis via VEGF and TGF-β signaling, respectively [2,4]. Consequently, pathologies involving cranial hemorrhage, vessel dilation, and vessel structural defects are common when pericytes are reduced or absent due to loss of key pericyte signalling pathways, such as Pdgfb [5–9]. Notch activity is also critical for emergence of perivascular mural cells, particularly smooth muscle cells [10–12]. Loss- and gain-of-function of Notch3 receptor in zebrafish revealed a role in brain pericyte proliferation [13]. In mice, pericyte loss due to Notch signaling deficiency leads to arteriovenous malformations [14].

files. Bulk and single-cell RNA-Sequencing data reported in this work are available through NCBI GEO (GSE232763). Code is provided in supplemental data files.

**Funding:** SJC was supported by Canadian Institutes of Health Research Project grant (PJT-153023). SA received fellowships from the Alberta Children's Hospital Research Insitute and the Cumming School of Medicine. CA received an Eyes High Doctoral Research Scholarship from the University of Calgary. PH was supported by a Canadian Institues of Health Resaerch Project grant (PJT-169113). JB was supported by Canadian Institutes of Health Research Project grant (PJT-4013940) SS was supported by CIHR Vanier, Alberta Innovates (AI), and Killam Doctoral Scholarships. EL was supported by an Alberta Children's Hospital Research Postdoctoral fellowship. QL was supported by a grant from the National Science and Engineering Research Council of Canada (RGPIN-2017-04860). The funders had no role in study design, data collection and analysis, decision to publish, or preparation of the manuscript.

**Competing interests:** The authors have declared that no competing interests exist.

**Abbreviations:** BFP, blue fluorescent protein; CtA, central artery; dpf, days postfertilization; GOF, gain-of-function; HCR, hybridization chain reaction; MZ, maternal zygotic; scRNAseq, single-cell RNA sequencing; UMAP, Uniform Manifold Approximation and Projection.

Owing to its critical function in vascular homeostasis, pericyte development has received much attention. Quail-chick chimeras showed that pericytes of the forebrain are neural crest derived while aortic pericytes originate from Pax1+ and FoxC2+ sclerotome [15,16]. Mesodermal and neural crest origins of pericytes have also been shown in the zebrafish, where pericytes of the anterior midbrain originate from neural crest and those in the hindbrain and trunk are derived from the paraxial mesoderm [17]. Transcriptional and signalling pathways that promote pericyte differentiation include the forkhead box transcription factors FoxC1 and FoxF2 that are required for brain pericyte differentiation and blood–brain barrier maintenance [18,19]. Furthermore, mice and zebrafish lacking FoxC1 and FoxF2 show cerebral hemorrhages [18–20]. In line with this, humans with risk loci near *FOXF2* are more susceptible to stroke and cerebral small vessel disease [21].

While pathways promoting pericyte differentiation have been discovered, the initial signals triggering the convergent differentiation of brain pericyte precursors from 2 different germ layers are unknown. Here, we describe the role of a homeobox transcription factor Nkx3.1, which is required in pericyte precursors originating in both neural crest and paraxial mesoderm. *nkx3.1*$^{-/-}$ mutants exhibit fewer pericytes on brain vessels and brain hemorrhage. Single-cell sequencing on *nkx3.1*-lineage cells reveals that pericyte precursors are marked by the transcription factors *tbx18* and *foxf2a* among other genes. Furthermore, we show that chemokine ligand *cxcl12b/sdf1* is expressed in pericyte precursors and functions downstream of *nkx3.1* during pericyte development. Taken together, our study defines a previously unknown pericyte precursor population and a novel Nkx3.1-Cxcl12b cascade during pericyte development.

## Results

### Brain pericytes originate from a lineage marked by *nkx3.1*

An early marker of the zebrafish sclerotome is the transcription factor, *nkx3.1*. Trunk pericytes are derived from *nkx3.1*-expressing sclerotome precursors, although *nkx3.1* is down-regulated when trunk pericytes differentiate [16,22]. Precursor markers for brain pericytes have not yet been identified. Brain pericytes form from 2 germ layers (neural crest and mesoderm), and an understanding of the convergent genetic program to differentiate cells from different origins into pericytes is also lacking. *nkx3.1* is expressed in the ventral head mesenchyme and trunk of the developing embryo at 16 and 30 hpf, as shown by in situ hybridization (Fig 1A and 1B, arrowheads). *nkx3.1* expression in ventral head mesenchyme is still present at 30 hpf but greatly reduced. It is undetectable by 48 hpf (Fig 1C). Expression of *nkx3.1* occurs far earlier than that of the pericyte marker pdgfrβ, first expressed at 48 hpf in the basilar artery [13]. To determine whether brain pericytes originate from *nkx3.1* lineage, we made use of the transgenic lines *TgBAC(nkx3.1:Gal4)*$^{ca101}$ and *Tg(UAS:NTR-mCherry)*$^{c264}$ to raise *nkx3.1:Gal4; UAS:Nitroreductase-mCherry*, hereafter known as *nkx3.1*$^{NTR-mcherry}$ embryos where *nkx3.1* lineage cells are labelled with mCherry. mCherry perdures after native *nkx3.1* mRNA is down-regulated, allowing us to track cell lineage beyond the time that *nkx3.1* mRNA is normally expressed [22–24]. At 4 days postfertilization (dpf), we observe *nkx3.1*$^{NTR-mcherry}$ cells in the perivascular zone surrounding endothelial cells (Fig 1D–1E', arrowheads). These *nkx3.1*-lineage cells show a pericyte-like morphology with a round soma and processes that wrap around endothelial cells (Fig 1E–1F', arrowheads). Furthermore, we found a complete overlap in expression between *nkx3.1*$^{NTR-mcherry}$ and a transgenic reporter of pericytes, *TgBAC(pdgfrb: GFP)*$^{ca41}$ at 75 hpf (Fig 1G–1I). This confirms that *nkx3.1*$^{NTR-mcherry}$ perivascular cells in the brain are pericytes at 75 hpf. Since *nkx3.1* is expressed prior to *pdgfrβ*, but their later expression is completely overlapping, these data suggest that *nkx3.1* is expressed in pericyte

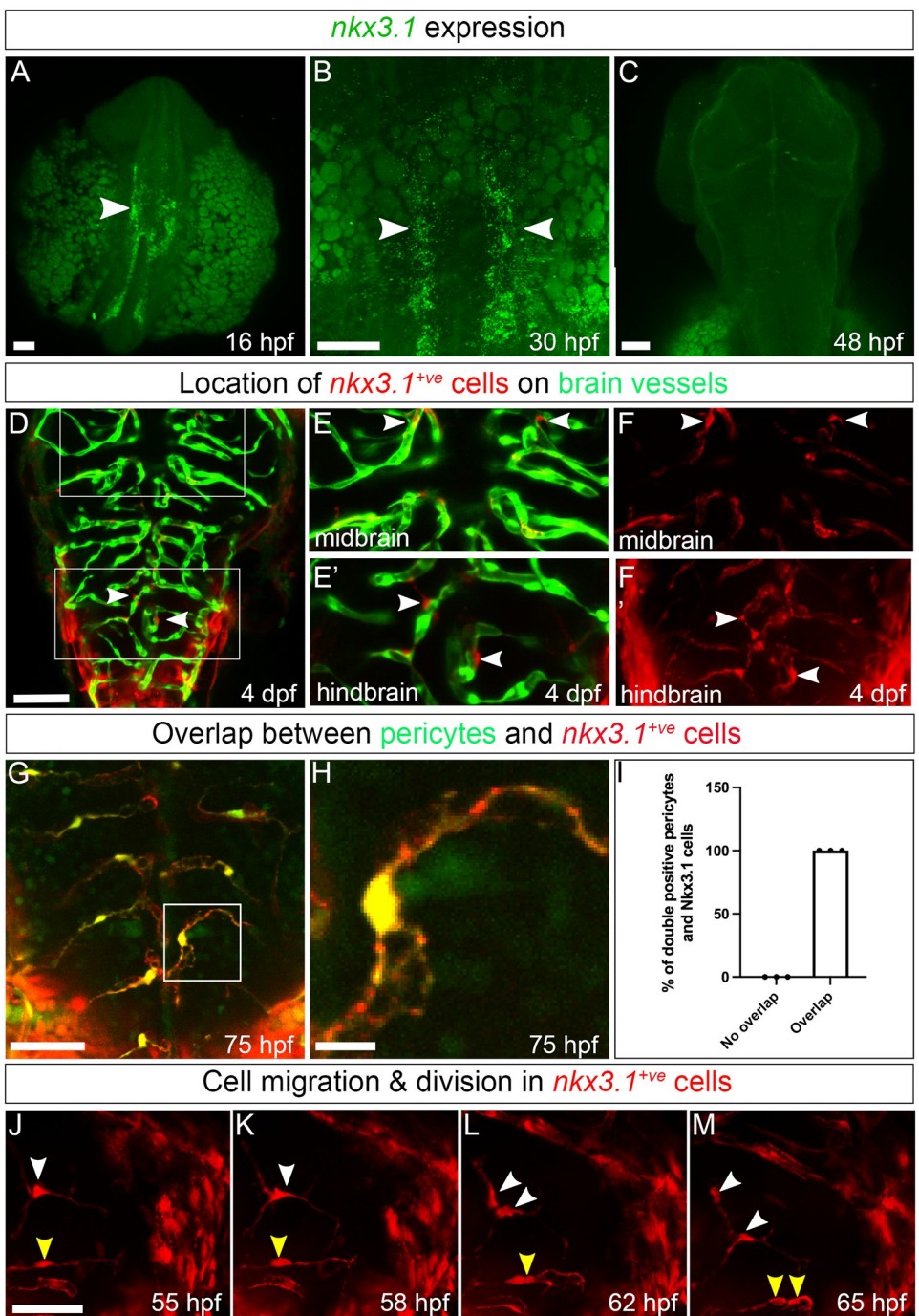

**Fig 1.** *nkx3.1* **expression patterns and cell behavior.** All images are captured dorsally and the anterior (A) and posterior (P) axis is marked. (**A**-**C**) Expression of *nkx3.1* by HCR in situ hybridization. (**A**) At 16 hpf, *nkx3.1* is expressed in the hindbrain and anterior trunk. Arrowheads mark *nkx3.1* expression. (**B**) At 30 hpf, *nkx3.1* is expressed in the posterior head and trunk. (**C**) At 48 hpf, *nkx3.1* expression is not detectable. (**D**-**E'**) *nkx3.1*^NTR-mcherry cells (red) are adjacent to endothelium (green; *Tg(flk:GFP)*) in brain vessels at 4 dpf. Pericytes in midbrain (arrowheads, **E**, **F**) and hindbrain (arrowheads, **E'**, **F'**) are denoted in dual channel (**E**, **E'**) and single-channel pericyte (**F**, **F'**) images. (**G**) Brain pericytes labelled by *TgBAC(pdgfrβ:GFP)* coexpress *nkx3.1*^NTR-mcherry 75 hpf. (**H**) Enlargement of an individual brain pericyte marked by a square in G. (**I**) Quantification brain pericytes coexpressing *nkx3.1* at 75 hpf (*N* = 3 experiments and 30 embryos). (**J**-**M**) Single images from time-lapse of *nkx3.1*^NTR-mcherry cells in the midbrain. White and yellow arrowheads track individual cells that migrate and divide with time. Scale bar in all images is 50 μm.

precursors. To characterize the cellular behavior of *nkx3.1*[NTR-mcherry] perivascular cells, we imaged labelled cells in the embryonic brain from 55 to 65 hpf. Time-lapse reveals that *nkx3.1*[NTR-mcherry] perivascular cells migrate and proliferate on blood vessels (Fig 1J–1M, white and yellow arrowheads, S1 Movie), similar to previously described cellular behavior of pericytes [17].

Previous lineage-tracing data suggest that pericytes in the zebrafish hindbrain originate from mesoderm and that midbrain pericytes originate from neural crest [17]. Pericytes in both hindbrain and midbrain express the *nkx3.1* transgene (Fig 1D). We next used lineage tracing of mesoderm and/or neural crest progenitors to test whether *nkx3.1*-expressing cells arose from one or both germ layers. We used Cre drivers for mesoderm *Tg(tbx6:cre;myl7:GFP)* or neural crest *Tg(sox10:cre;myl7:GFP)* together with a floxed reporter *Tg(loxp-stop-loxp-H2B-GFP)* to lineage label mesodermal or neural crest progeny, respectively. We crossed these fish with an *nkx3.1* reporter *TgBAC(nkx3.1:Gal4)*, pericyte reporter *TgBAC(pdgfrb:Gal4FF)*, or endothelial reporter *Tg(kdrl:mCherry)*. This strategy labels cells expressing *pdgfrβ*, *nkx3.1*, or *kdrl* in red and mesodermal or neural crest derivatives as green (depending on the experiment). We imaged double positive cells to identify their lineage and observe that both mesoderm and neural crest progenitors contribute to both hindbrain and midbrain pericytes by lineage tracing either pdgfrβ or *nkx3.1* (S1 Fig).

Taken together, our data show that *pdgfrβ*-expressing brain pericytes originate from an *nkx3.1* positive precursor, which, in turn, is generated from *tbx6 and sox10* expressing mesodermal and neural crest progenitors. *nkx3.1* is, therefore, a transcription factor expressed in precursors of brain pericytes but not in mature brain pericytes.

## Nkx3.1 function is required for brain pericyte development

To determine whether Nkx3.1 function is necessary for brain pericyte development, we made *nkx3.1* mutant zebrafish using CRISPR-Cas9. *nkx3.1*[ca116] mutants have a 13-bp deletion in exon 2 that is predicted to lead to premature stop prior to the homeobox domain (S2 Fig). *nkx3.1*[−/−] mutant embryos show no gross morphological defects and adults are homozygous viable, although with reduced life span of approximately 1 year. At 75 hpf, we observed no difference in total pericyte number (sum of mid- and hindbrain pericytes) between *nkx3.1*[−/−] mutants and their heterozygous and wild-type siblings (S3 Fig); however, a previous study showed that *nkx3.1* transcripts are maternally contributed to the developing embryo [25]. To remove this maternal contribution, we crossed *nkx3.1*[+/−] males with *nkx3.1*[−/−] females. Maternal zygotic (MZ) *nkx3.1*[−/−] embryos show a strong phenotype including brain hemorrhage and hydrocephalus at 52 hpf as compared to controls (Fig 2A and 2B). The average percentage of MZ *nkx3.1*[−/−] embryos exhibiting brain hemorrhage at 52 hpf (54%) was significantly higher as compared to *nkx3.1*[+/−] siblings (12.5%; Fig 2C). Not surprisingly, given altered hemodynamics after hemorrhage, central artery (CtA) vessel diameter was decreased from 5.9 to 5.4 µm on average (Fig 2D).

Importantly, MZ *nkx3.1*[−/−] showed significantly fewer pericytes at 75 hpf, as observed by confocal microscopy (Fig 2E–2G). This is consistent with the brain hemorrhage phenotype, which is a reported consequence of lack of pericytes [26]. Furthermore, the density of pericytes (defined as the number of pericytes divided by the length of the vessel network) is also reduced (Fig 2H). The total length of the central arteries (CtA network length) is unchanged, suggesting that the endothelial patterning is unaffected (S4 Fig) These phenotypes are maintained at into early larval stages; both pericyte number and pericyte density are also decreased at 5 dpf in *nkx3.1* mutants, while CtA network length is unchanged (S4 and S5 Figs). All work further, therefore, used MZ *nkx3.1* mutants.

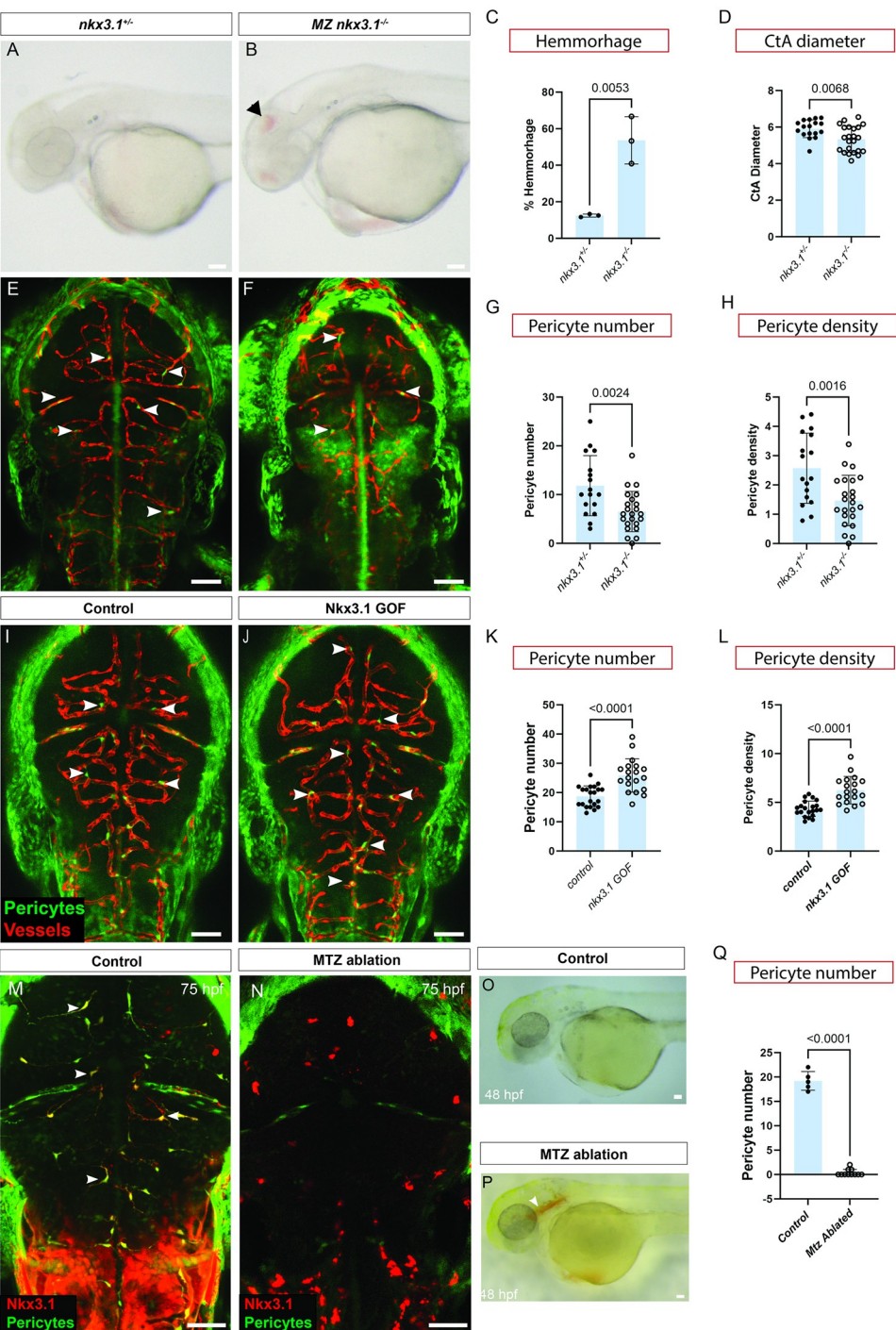

**Fig 2. Nkx3.1 function is required to regulate brain pericyte numbers.** Lateral view of control *nkx3.1*<sup></sup>$nkx3.1^{+/-}$ (**A**) and $nkx3.1^{-/-}$ MZ (**B**) mutants showing brain hemorrhage (arrowhead) at 52 hpf, and quantification (**C**; $N = 3$, proportions of hemorrhage). (**D**) $nkx3.1^{-/-}$ have decreased CtA vessel diameter in comparison to $nkx3.1^{+/-}$ hets. Dorsal images of $nkx3.1^{+/-}$ hets (**E**) and $nkx3.1^{-/-}$ (**F**) embryos expressing *Tg(pdgfrβ:GFP)* and *Tg(kdrl:mCherry)* showing fewer brain pericytes (arrows) in mutants at 75 hpf. Quantification of (**G**) decreased pericyte number and (**H**), decreased pericyte density (defined as the number of pericytes divided by the length of the vessel network) in mutants. In comparison to control wild-type embryos (**I**, *Tg(hsp70l:tBFP)*), *nkx3.1* gain-of-function (GOF) embryos expressing *Tg(hsp70l:tBFP-2a-nkx3.1))* show more pericytes (**J**, arrowheads) ($n = 20$ control and 19 GOF)) as quantified (**K**). Pericyte density is also increased (**L**). Dorsal views of embryos labelled with *TgBAC(nkx3.1:Gal4)* and *Tg(UAS:NTR-mCherry)* under fluorescence (**M, N**) or under brightfield (**O, P**) that are untreated (**M, O**) or treated with

metronidazole (**N**, **P**) to ablate *nkx3.1*-expressing cells. Ablated embryos show brain hemorrhage (**P**, arrowhead) at 48 hpf (**P**). (**Q**) Quantification of pericyte number in ablated embryos. Statistical significance was calculated using the Student *t* test (*n* = 5 wild type and 11 *nkx3.1* mutants). Scale bars are 50 μm. The data underlying this figure can be found in S3 Table.

To test the sufficiency of Nkx3.1 for pericyte development, we made an *nkx3.1* gain-of-function (GOF) transgenic line *Tg(hsp70l:tBFP-2a-nkx3.1)* expressing blue fluorescent protein (BFP) fused with *nkx3.1*. *Tg(hsp70l:tBFP)* as a control. Overexpression of *nkx3.1* using heat shock from 29 to 30 hpf, when first brain pericytes are emerging, results in significantly more brain pericytes at 75 hpf, compared to expression of tBFP alone (Fig 2I–2K). Brain vessels of *nkx3.1* GOF embryos appear grossly morphologically normal. Furthermore, the density of pericytes on vessels is increased in *nkx3.1 GOF*, suggesting the increased number of pericytes is spread more tightly on the same length of vessel (Fig 2L). Thus, *nkx3.1* is both necessary and sufficient for brain pericyte development.

As a second method to test the necessity of *nkx3.1* precursors, we made used the *nkx3.1^{NTR-mcherry}* transgenic line to ablate *nkx3.1^{NTR-mcherry}* cells by treating with 5 mM metronidazole from 24 to 48 hpf, a time window during which brain pericyte differentiation is occurring [17]. Consistent with the MZ *nkx3.1^{−/−}* phenotype, ablation of *nkx3.1^{NTR-mcherry}* positive cells in transgenic embryos treated with metronidazole showed no pericytes at 75 hpf. *nkx3.1* thus appears to be expressed in all brain pericyte precursors as no pericytes remained after ablation (Fig 2M, 2N and 2Q). We also observed brain hemorrhage at 48 hpf after ablation (Fig 2O and 2P).

## Pericyte precursors share markers with fibroblasts

Although pericytes derive from *tbx6*-positive mesodermal cells and *sox10*-positive neural crest cells, intermediate progenitors have not yet been defined. We have shown that *nkx3.1*-expressing cells differentiate into pericytes and that *nkx3.1* is a marker of pericyte progenitors. Therefore, *nkx3.1* positive cells sampled at a stage prior to pericyte differentiation are a unique population to interrogate the gene expression profile of pericyte precursors. Since not all *nkx3.1* positive cells become brain pericytes (i.e., some become fibroblasts derived from sclerotome cells of the trunk and other lineages), embryos expressing the *nkx3.1^{NTR-mcherry}* transgene were dissociated from wild-type 30 hpf zebrafish embryos and mCherry-positive FACs sorted cells were subjected to single-cell RNA sequencing (scRNAseq; Fig 3A and 3B).

A total of 3,359 cells obtained from 2 biologically independent samples passed quality control. Using the Uniform Manifold Approximation and Projection (UMAP) algorithm in Seurat [27], we detected 13 different populations (Fig 3A and S1 Table). We identified unique markers in each cluster (Figs 3D and S8–S11). Cluster assignment used comparisons with published scRNAseq data (S1 Table) [28–32]. Canonical fibroblast markers (*col1a1a*, *col1a1b*, *pdgfrα*, *col5a1*, *mmp2*; S7 Fig) are expressed by a group of connected clusters, including 3 more differentiated clusters (Fb-A, Fb-b, and Fb-V) and 1 cluster that expresses both fibroblast and mitotic markers (Progenitor-like; Prog-2). Additional clusters not followed up here express mesoderm and heart, central nervous system (CNS), endothelial, basal fin (fibroblasts of the epithelial layer), or neutrophil markers.

To understand the relationships between clusters, we used RNA velocity, which measures transient transcriptional dynamics to infer the differentiation process, from analysis of RNA splicing information in sequencing data. We subclustered 2,120 cells in 5 clusters (Prog-1/2 and Fb-A/B/V; Fig 3C). Using RNA velocity, we find that flow direction is consistent with 2

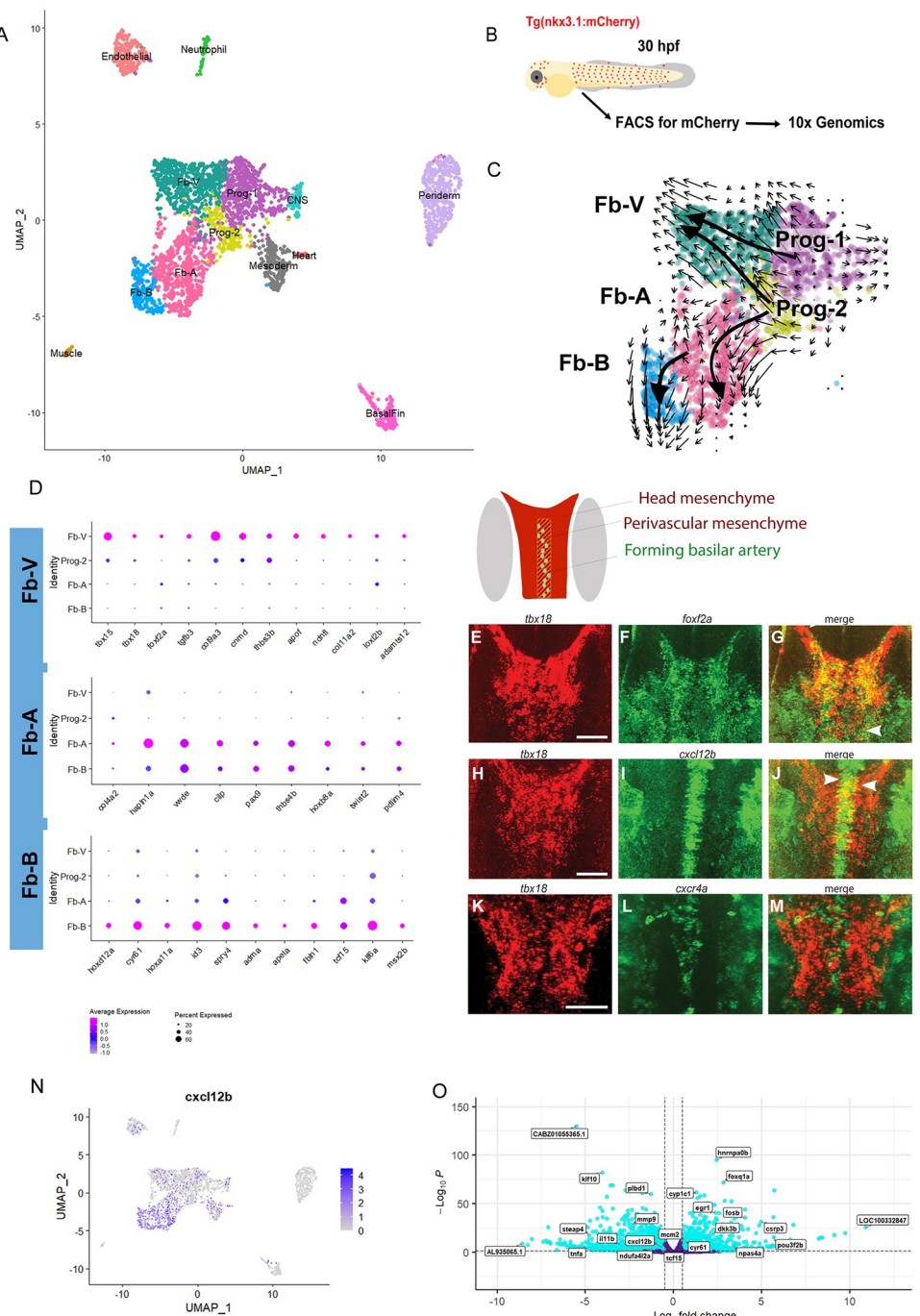

**Fig 3. Next-generation sequencing analysis of *nkx3.1*^NTR-mcherry^ and *nkx3.1*^−/−^ embryos at 30 hpf.** (**A**) Single-cell clusters of *nkx3.1*^NTR-mcherry^ embryos at 30 hpf. (**B**) Schematic showing workflow for single-cell sequencing of *nkx3.1*^NTR-mcherry^ cells. (**C**) RNA-velocity analysis of subclustered *nkx3.1*^NTR-mcherry^ cells of the Progenitor and FB-V, FB-A, and Fb-B clusters. (**D**) Dot plots showing key marker genes of FB-V, FB-A, and Fb-B clusters. (**E-M**) HCR expression analysis of Fb-V genes at 36 hpf with the schematic showing relative locations of the ventral head mesenchyme and the forming basilar artery in a dorsal schematic of the whole zebrafish brain (grey marks the position of the eyes). (**E-M**) HCR fluorescent in situ hybridization imaged by confocal showing substacks in the region of the forming basilar artery and precursor area. (**E-G**) *tbx18* (red) and *foxf2a* (green) show expression overlap at 36 hpf in the ventral head (yellow, arrowheads). (**H-J**) *tbx18* (red) and *cxcl12b* (green) show expression overlap at 36 hpf in the ventral head (yellow, arrowheads). (**K-M**) *tbx18* (red) is expressed in the perivascular space surrounding the *cxcr4a* (green) expressing basilar artery in the ventral head at 36 hpf. (**N**) *cxcl12b* feature plot showing its expression across clusters including Fb-V. (**O**) Bulk sequencing volcano plot of *nkx3.1*^−/−^ embryos at 30 hpf. Scale bar is 50 μm.

progenitor pools feeding into the fibroblast clusters, with the Progenitor 1 (Prog-1) cluster feeding into Fb-V and Progenitor 2 (Prog-2) feeding into all 3 fibroblast-like clusters (Fb-A, Fb-B, and Fb-V). Fb-B is a smaller cluster (6% of sorted *nkx3.1* cells) that appears to arise from the Fb-A fibroblast.

Based on a gene expression profile overlapping with some pericyte markers, the Fb-V cluster most likely represents a pericyte progenitor cluster. Canonical pericyte markers like *pdgfrβ*, *cspg4*, and *notch3* are present in scattered cells in these clusters at this time point but not yet enriched in Fb-V (S11 Fig). However, additional 3 pericyte markers, *tbx18*, *foxf2*, and *cxcl12b*, are enriched in Fb-V and mark *nkx3.1*-positive pericyte precursors, at this early stage prior to when canonical pericyte markers like *pdgfrβ* are expressed.

*tbx18* is expressed in the embryonic zebrafish head paraxial mesoderm [33] and has enriched expression in adult mouse brain mural cells. A lineage trace of mouse *Tbx18* shows expression in both adult mouse pericytes and vascular smooth muscle cells [34,35]. *foxf2* is also enriched in the Fb-V cluster. We have previously shown that *foxf2a* and *foxf2b* are expressed in zebrafish brain pericytes and are essential for generating the proper number of brain pericytes [20,36]. Furthermore, mouse *Foxf2* is enriched in and critical for mouse brain pericyte formation [21,31]. A third gene we explore in the Fb-V cluster is *cxcl12b* (Fig 3N). Vascular mural cells associated with the zebrafish coronary arteries and caudal fin vessels express *cxcl12b* [37,38]. In addition, mural cells of the mouse and human lung tissue express *Cxcl12* [39]. Since our data are collected at a stage where there is scant information about the differentiating pericyte gene expression profile from any species, our data will be very useful for further studies (S1 Table).

To validate expression of genes of interest from the Fb-V cluster, we used in situ hybridization at 36 hpf when the first pericytes are associating with developing brain vessels. We find that *tbx18* is expressed in the perivascular space around the *cxcr4a*+ basilar artery (Fig 3K–3M), where the first brain pericytes attach [17]. We also detect *foxf2a* and *cxcl12b* expression in *tbx18*+ cells (Fig 3E–3J). These data validate single-cell sequencing results and confirm the presence of a perivascular cell type that coexpresses *tbx18*, *foxf2a*, and *cxcl12b* around the forming basilar artery, the first site of pericyte attachment [17].

The Fb-A and Fb-B clusters are diverging from Fb-V and express genes reminiscent of sclerotome, including *pax9* and *twist2*, while the Fb-B cluster is enriched for fibroblast markers such as *cyr61*, and the *tcf15/paraxis*, involved in trunk mesoderm development. Both clusters express *thrombospondin 4b (thbs4b)*. Since Fb-B flows away from Fb-A, which flows away from Prog-2, this suggests that this lineage is differentiating into traditional fibroblasts that will go on to assume many different lineages, including those of the zebrafish trunk [23,24].

## Loss of *nkx3.1* leads to transcriptional reduction of the chemokine *cxcl12*

To determine which genes within *nkx3.1*-expressing cells are important for their differentiation, we took advantage of *nkx3.1*$^{ca116}$ genetic mutant fish. At the identical stage to the scRNA-seq (30 hpf), we sampled *nkx3.1* wild-type and MZ *nkx3.1* mutant embryos using bulk RNA sequencing. This revealed that 2,788 genes were down-regulated, and 2,080 genes up-regulated at 30 hpf (S2 Table). Among the genes down-regulated at 30 hpf is *tcf15*, the pericyte marker *ndufa4l2a*, and chemokine *cxcl12b* (Fig 3O). Since we showed that *cxcl12b* is expressed in the pericyte progenitor cluster (Fb-V), we next interrogated the role of Cxcl12b in pericyte development.

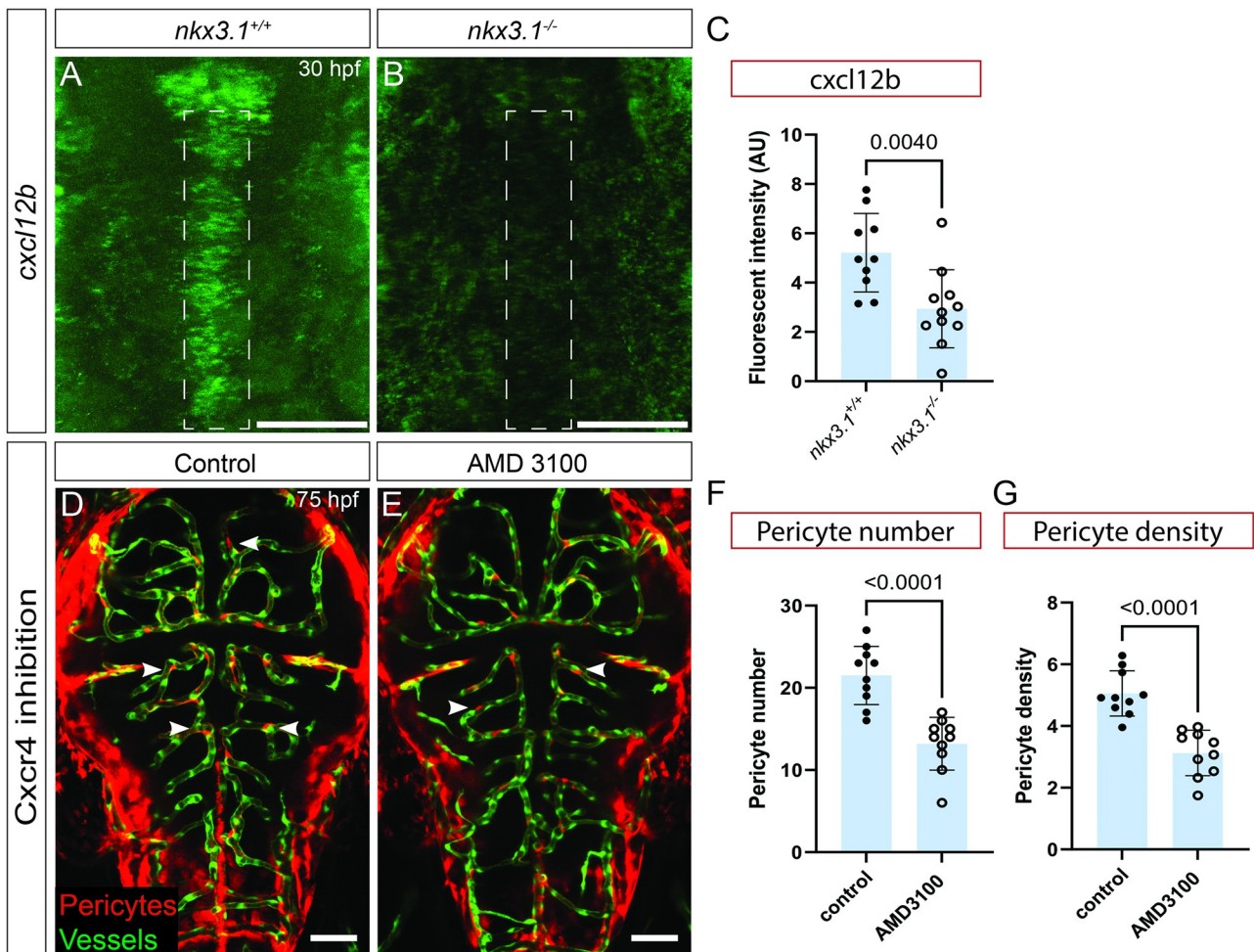

**Fig 4. *cxcl12b* is regulated by *nkx3.1*; loss of Cxcr4 signaling reduces brain pericytes.** All embryos were imaged dorsally in the head region. *cxcl12b* mRNA expression is reduced in the embryonic head of *nkx3.1^{-/-}* mutants (**B**) as compared to wild-type controls (**A**) at 30 hpf as quantified by fluorescent intensity (**C**) of regions marked by dotted lines in A and B. (*n* = 10 wild-type and 10 mutant embryos). Inhibition of Cxcr4 from 24–75 hpf using AMD 3100 leads to reduced brain pericytes in treated (**E**) vs. untreated (**D**) embryos (at 75 hpf (*n* = 10)). Pericytes (red cells, white arrows) are labeled by *TgBAC(pdgfrβ:Gal4)*; *Tg(UAS:NTR-mCherry)*. Brain vessels in D and E are labeled by *Tg(flk:GFP; green)*. Quantification of (**F**) pericyte number and (**G**) pericyte density. (*n* = 10 wild type and 10 mutants). Statistical significance was calculated using the Student *t* test. Scale bar is 50 µm. The data underlying this figure can be found in S3 Table.

## Cxcl12b signaling regulates brain pericyte number

Using in situ hybridization at 16 hpf and 24 hpf, we find that *cxcl12b* and *nkx3.1* are coexpressed in the ventral head mesenchyme of the developing embryo (S12 Fig). We next tested expression of *cxcl12b* in *nkx3.1^{-/-}* mutants. At 30 hpf, *cxcl12b* is strongly down-regulated in the head of MZ *nkx3.1^{-/-}* mutants at a location where the first pericytes will attach to the basilar artery (Fig 4A–4C). To test the role of Cxcr4-Cxcl12 signaling in pericyte development, we used AMD 3100, an inhibitor of the Cxcl12 receptor, Cxcr4. The number of brain pericytes and brain pericyte density is significantly reduced after treatment of zebrafish embryos with 100 µM AMD 3100 from 24 to 75 hpf (Fig 4D–4G), suggesting requirement of Cxcl12b signaling in pericyte development. The CtA network length is unchanged (S4 Fig). Consistent with this finding, GOF induced by mRNA injection of 30 pg *cxcl12b* mRNA not only increases

brain pericyte number in wild-type embryos but also increases brain pericyte number in MZ *nkx3.1* mutants at 75 hpf. However, GOF *cxcl12b* expression does not decrease hemorrhage when expressed in *nkx3.1* mutants (S13 Fig). We note that there is no overlap in cxcr4a and cxcl12b expression at 36 hpf (S14 Fig), suggesting that cxcl12b signals nonautonomously to cxcr4a expressing cells.

Supporting our results, single-cell sequencing of zebrafish at multiple stages as reported in the DanioCell database [32] shows expression of *nkx3.1*, *foxf2a* and *cxcl12b* at the 24 to 34 and 36 to 48 hpf at a time when *pdgfrβ* is only weakly expressed (S15 Fig). This independent dataset strongly supports the coexpression of these 3 genes in the pericyte lineage prior to definitive pericyte marker expression.

Taken together, our data show that Cxcl12b is coexpressed with and required downstream of Nkx3.1 in pericyte development.

## Cxcl12b function is required in *nkx3.1*$^{+ve}$ precursors for brain pericyte development

We next tested whether Cxcl12b is required in *nkx3.1*$^{+ve}$ precursors or Pdgfrβ$^{+ve}$ pericytes or in both cell types. We used transgenic overexpression of *cxcl12b* in *nkx3.1*$^{-/-}$ mutants by constructing *Tg(UAS:cxcl12b;cryaa:mKate)* and crossing to Gal4 drivers in either the *nkx3.1*$^{+ve}$ precursor lineage or Pdgfrβ$^{+ve}$ pericytes using *nkx3.1*$^{-/-}$;*TgBAC(nkx3.1:Gal4)*$^{ca101}$ or *nkx3.1*$^{-/-}$;*TgBAC(pdgfrb:Gal4FF)*$^{ca42}$. We scored the number of brain pericytes in mutants carrying both transgenes. We find that *nkx3.1*$^{-/-}$ mutant pericyte numbers are increased by *nkx3.1*-driven *cxcl12b* overexpression (Fig 5A–5C) but not by *pdgfrβ*-driven *cxcl12b* overexpression at 75 hpf (Fig 5E–5G). Brain pericyte density is also increased when *cxcl12b* is expressed under the *nkx3.1* driver (Fig 5D). Agreeing with the mRNA overexpression data, these experiments confirm the requirement for Cxcl12b in an early stage of pericyte differentiation in the *nkx3.1* pericyte precursor population but not in later pericyte development (*pdgfrβ*).

## Discussion

Little is known about the essential factors that contribute to the developmental journey of a differentiated pericyte. Most studies focus either on the upstream lineage origins from mesodermal or neural crest precursors, or on downstream genes important for differentiated pericytes, but the intermediate genetic factors driving lineage differentiation are less well known. Here, we show that *nkx3.1* is a vital intermediate gene in the differentiation journey of a pericyte (model figure; Fig 6). Using a transgenic reporter of *nkx3.1* expression, we show that it precedes Pdgfrβ expression in the pericyte lineage. Cells in the brain that express *nkx3.1* become *pdgfrβ*-expressing pericytes. Critically, *nkx3.1* is required for brain pericytes of both mesodermal and neural crest origin, suggesting that it is a gene that either unifies the convergent pericyte differentiation program from different lineages or is present in the newly unified population. Beyond the brain, previous work shows that *nkx3.1* is expressed in perivascular fibroblasts precursors of the trunk pericyte lineage [22]. It is likely that *nkx3.1* is important for pericyte and fibroblast development in other areas of the embryo. Homozygous *nkx3.1* mutants are viable and fertile with reduced life span, suggesting that progeny of embryonic *nkx3.1*-expressing brain pericytes, trunk pericytes, and/or fibroblasts likely contribute to progressive postembryonic phenotypes. We show that the absence of *nkx3.1*-expressing cells results in a severe reduction in pericyte number. Loss of *nkx3.1* and gain of *nkx3.1* show that *nkx3.1* is both necessary and sufficient for modulating brain pericyte number. RNAseq and scRNAseq reveal the transcriptome of *nkx3.1*-expressing pericyte precursors as similar to

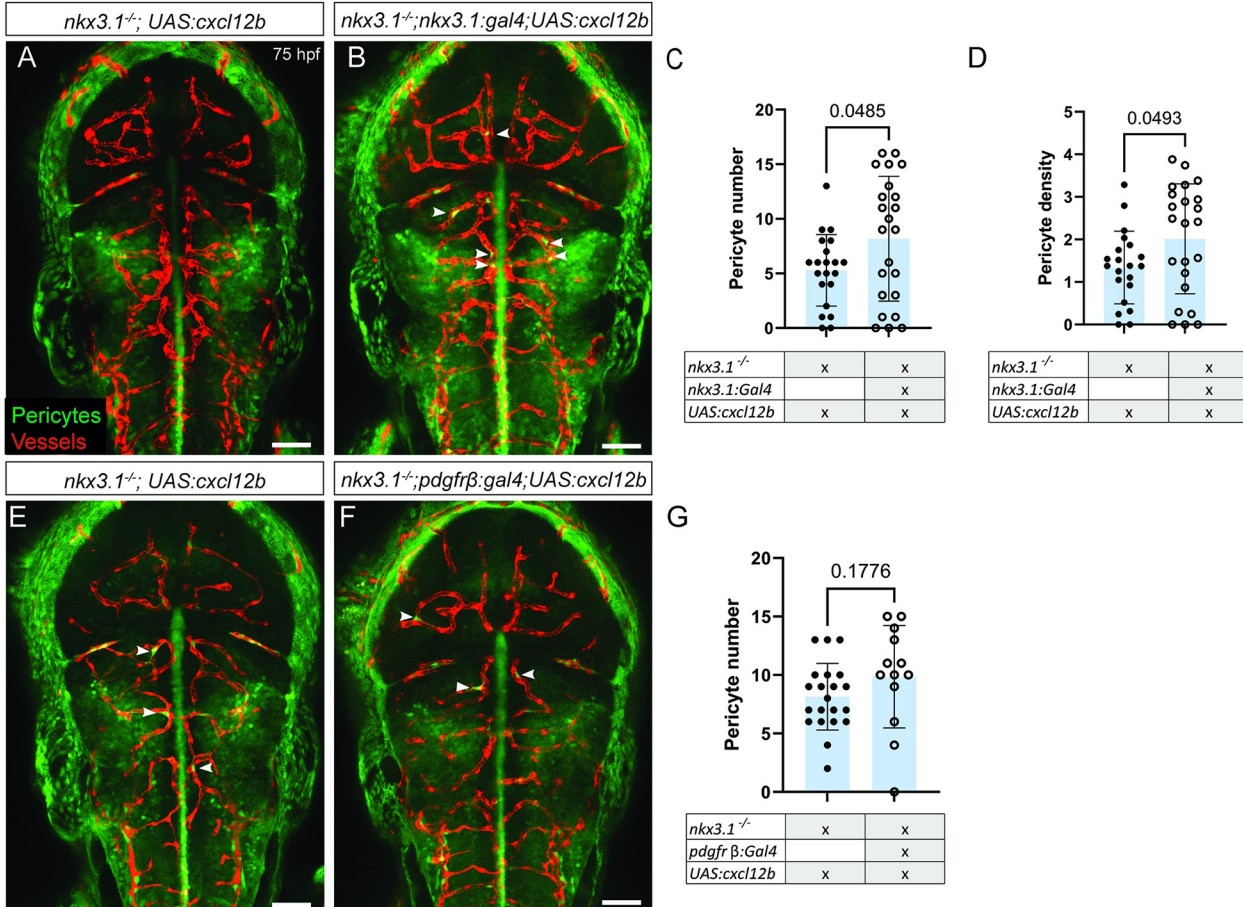

**Fig 5. Reexpression of Cxcl12b increases pericyte numbers in *nkx3.1*$^{-/-}$.** All embryos were imaged dorsally in the head region. In comparison to a nkx3.1$^{-/-}$ mutant without a Gal4 driver (**A**), expressing UAS:*cxcl12b* under the *nkx3.1* Gal4 driver *TgBAC(nkx3.1:Gal4)* increases pericyte numbers (**B**, **C**) and pericyte density (**D**) in *nkx3.1* mutants at 75 hpf (*n* = 21 mutants without nkx3.1Gal4 and 22 with Gal4). In comparison to a nkx3.1$^{-/-}$ mutant without a Gal4 driver (**E**), expressing UAS:*cxcl12b* under the under the *pdgfrβ* Gal4 driver *TgBAC(pdgfrβ:Gal4)* does not change pericyte numbers at 75 hpf (**F**, **G**; *n* = 16 mutants without the pdgfrβ:Gal4, and 8 with pdgfrβ:Gal4). Arrowheads mark example brain pericytes (green) labeled by *TgBAC(pdgfrβ:GFP)*. Brain vessels (red) are labeled by *Tg(kdrl:mCherry)*. Statistical significance was calculated using the Student *t* test. Scale bar is 50 μm. The data underlying this figure can be found in S3 Table.

fibroblasts and identify the chemokine Cxcl12 as a critical factor for brain pericyte differentiation, whose expression is controlled by Nkx3.1.

Brain pericytes are unusual as they arise from 2 distinct lineages, mesoderm and neural crest in both fish and mouse. No functional differences have been noted between pericytes from different origins. Using lineage tracing, we show that pericytes derived from both paraxial mesoderm (*tbx6*) and neural crest (*sox10*) express *nkx3.1* and that pericytes derived from both mesoderm and neural crest are present in both the midbrain and hindbrain. Previous work has suggested a more compartmentalized contribution in fish where mesoderm contributes to hindbrain pericytes and neural crest contributes to midbrain pericytes [17]. This may have occurred due to incomplete sampling in these difficult experiments, as similar reagents were used. The rarity of recombination events limits precise quantification of mesodermal and neural crest contribution, but contributions from both mesodermal (myeloid) and neural crest lineages to brain pericytes are also observed in mice [40].

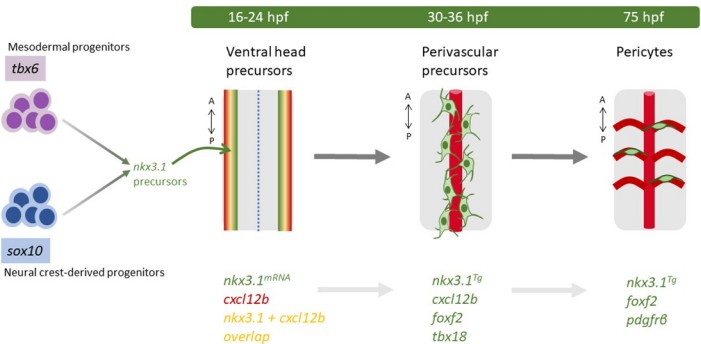

**Fig 6. nkx3.1 is essential in pericyte precursors.** Model of *nkx3.1* in pericyte differentiation. *nkx3.1* is expressed in cells of both mesodermal and neural crest origin as determined by lineage analysis. All embryo schematics show a dorsal view of the hindbrain. At 16 and 24 hpf, nkx3.*1* and *cxcl12b* coexpressing precursors are located ventrolaterally. At 36 hpf, nkx3.*1*, *cxcl12b*, *tbx18*, and *foxf2* coexpressing precursors surround the developing basilar artery (BA). By 75 hpf, pdgfrβ-expressing pericytes derived from *nkx3.1* precursors have migrated to the central arteries of the brain. Key genetic markers are indicated.

Ablation of *nkx3.1*-expressing cells or loss of the *nkx3.1* gene within these cells leads to an identical phenotype. Loss of *nkx3.1*-expressing cells or of *nkx3.1* leads to phenotypes associated with pericyte disruption, including brain hemorrhage and reduced pericyte number [20,26,41], suggesting a critical role of Nkx3.1 in brain pericyte development. The level of pericyte loss is similar to that reported for both *foxf2* and *notch3* knockout fish, suggesting that all 3 genes are key players in pericyte differentiation [13,20], potentially with some redundancy as *pdgfrβ* knockout zebrafish have no pericytes, while all the models of transcription factor loss (*nkx3.1, foxf2a, foxf2b*) show reduced pericytes. Notch 3 GOF increases pericyte numbers, and we show here that gain of Nkx3.1 leads to an increase in pericyte number. However, there was no change in *notch3* expression in our RNA sequencing to indicate that there is a regulatory relationship, despite similar functions. Instead, Nkx3.1 and Notch3 may potentially act using similar downstream mechanisms [13].

Where and when *nkx3.1* acts in the pericyte differentiation cascade is an important question. We note that *nkx3.1* is an early embryonic gene, and by 48 hpf, its mRNA is undetectable; however, perdurance of the *nkx3.1*^NTR-mcherry^ transgenic allowed us to follow cells after the endogenous gene has turned off. Nothing is known of the Nkx3.1 protein and how long it may remain in an embryo; however, our data point to a specific early role in development. Incidentally, the first brain pericytes attach to the basilar artery by 36 hpf, and *nkx3.1* is expressed before this time.

To define the gene signature of this pericyte precursor population before pericyte markers are observed, we used single-cell sequencing of sorted *nkx3.1*^NTR-mCherry^ cells at 30 hpf. Analysis of single cells revealed *nkx3.1* contribution in 13 gene clusters, of which the majority of *nkx3.1* positive cells belong to 2 precursor clusters and 3 fibroblast-like clusters as defined by expression of pan-fibroblast markers such as *col1a1*, *col5a1*, and *pdgfrα* (S7 Fig) [31]. Interestingly, the Fb-V cluster is enriched in genes expressed in pericytes and/or crucial for their development, i.e., *tbx18* [35,42], *cxcl12b* [37–39], and *foxf2a* [19–21]. However, the Fb-V cluster lacks expression of "classical" differentiated pericyte markers, including *pdgfrβ* [17], *abcc9* [12,43], *kcnj8* [43], *ndufa4l2a*, and *kcne4* [44]. This indicates that Fb-V are fibroblast-like precursors and not differentiated cells. Differentiated pericytes are observed 2 days later in development. Of the 2 precursor clusters, Prog1 has extremely high expression of ribosomal protein subunits (30 *rps* (small ribosome) and 44 *rpl* (large ribosome) genes). Expression of *rps* and *rpl*

genes is very enriched in mural cells from early human development (GW15-18) in comparison to later development (GW20-23) [28], suggesting that this cluster represents early precursors. The second precursor cluster, Prog2, overlaps in expression profile with all 3 fibroblast clusters (A, B, and V). RNA velocity analysis suggests that Prog1 contributes to Fb-V and *nkx3.1*-expressing derivatives in the head, while Prog2 contributes to Fb-V, the 2 sclerotome fibroblast clusters Fb-A and Fb-B and heart mesoderm. RNA velocity also suggests that Fb-B is further differentiated from Fb-A. These 2 fibroblast clusters express classical markers of sclerotome, an expected major population of *nkx3.1*-expressing cells [22]. Although it has a similar gene expression profile, sclerotome is spatially separate from brain pericytes in the embryo and forms the trunk mesenchyme. Extensive characterization of the pericyte transcriptome by scRNAseq in the adult mouse, differentiated fish pericyte (5 dpf), and embryonic human are all from later developmental stages than the transcriptome that we determine here, and, therefore, our analysis represents an early snapshot of pericyte differentiation [28,44,45].

To understand functional targets of Nkx3.1 in pericyte differentiation, we undertook bulk RNA sequencing of *nkx3.1* mutants. Among the down-regulated genes, we found *ndufa4l2a* (a pericyte marker) [44] and *cxcl12b* are both down-regulated in $nkx3.1^{-/-}$ embryos at 30 hpf. This is intriguing as scRNAseq showed expression of chemokine *cxcl12b* in *nkx3.1*-expressing fibroblasts, including Fb-V. Additionally, Cxcl12-cxcr4 signalling is involved in bone marrow–derived pericyte differentiation [46], and Cxcl12 also plays a role in recruitment of vascular smooth muscle cells to the zebrafish aorta [47], suggesting it is a strong candidate for mediating effects downstream of nkx3.1. *nkx3.1* positive pericyte precursors coexpress *cxcl12b*, *tbx18*, *and foxf2a*, confirming our scRNAseq data. Functional small molecule inhibition of Cxcl12-Cxcr4 signaling significantly reduced pericyte numbers, and expressing *cxcl12b* under promoters for either *nkx3.1* (early precursors) or *pdgfrb* (more mature pericytes) showed that it was only able to increase pericyte numbers in *nkx3.1* mutants when expressed early (*nkx3.1* driver), but not once pericytes had differentiated to express (*pdgfrb* driver*)*. Taken together, this suggests that expression of *nkx3.1* in a pericyte precursor promotes the expression of the chemokine *cxcl12* and influences pericyte differentiation. We propose a model where Cxcl12 released by pericyte precursors binds to Cxcr4a expressed by the basilar artery. Based on the model for smooth muscle cell recruitment [47], endothelial cells, in turn, might produce the Pdgfb ligand to facilitate attachment of Pdgfrβ+ pericytes on the basilar artery [48]. Previous work suggesting that Pdgfb is attenuated with Cxcl12-Cxcr4 signalling inhibition supports our hypothesis [46].

Our study identifies a new player in pericyte differentiation, the transcription factor, Nkx3.1. Nkx3.1 is required in an intermediate precursor cell state that exists temporally downstream of germ layer (mesoderm or neural crest) specification and upstream of differentiated pericytes expressing canonical makers. The role of Nkx3.1 in brain pericyte precursors is transient and occurs in parallel to its role in trunk sclerotome, although these are distinct embryonic populations. We identify the key role of Nkx3.1 in promoting the proper number of pericytes to emerge on brain vessels to promote downstream vascular stability. We show that expression of *nkx3.1* is necessary and sufficient to modulate developmental pericyte number, although it does not result in complete loss of pericytes, suggesting partial redundancy. Defining the novel gene expression signature of *nkx3.1*-expressing pericyte precursors using scRNAseq opens new avenues for understanding pericyte differentiation. For instance, expression of 2 transcription factors in the Fb-V cluster (*foxf2a*, *tbx18*) are associated with pericytes in previous studies [20,21,34,35], although have poorly described roles, and regulatory changes in FOXF2 are associated with stroke in humans [20,49]. Future work focusing on identifying additional intermediate genes in pericyte differentiation is needed to illuminate the stepwise differentiation of pericytes from upstream precursors, and potential for regeneration in disease where pericytes are lost.

## Methods

### Zebrafish

All procedures were conducted in compliance with the Canadian Council on Animal Care, and ethical approval was granted by the University of Calgary Animal Care Committee (AC21-0189). All experiments included wild-type or vehicle-treated controls as a comparison group and developmental stages, n's, genotypes, and statistical outcomes are noted for each experiment. Embryos were maintained in E3 medium at 28˚C. For heat shock experiments, embryos were heated for 1 hour at 39˚C in a heating block.

The following published strains were used: *TgBAC(nkx3.1:Gal4)*$^{ca101}$ [23], *Tg(UAS:NTR-mCherry)*$^{c264}$ [50], *Tg(kdrl:GFP)*$^{la116}$ [51], *TgBAC(pdgfrβ:GFP)*$^{ca41}$ [3], *TgBAC(pdgfrβ:Gal4)*$^{ca42}$ [3], *Tg(kdrl:mCherry)*$^{ci5}$ [52].

The following strains were generated for this manuscript: *Tg(hsp70l:tBFP)*$^{ca90}$, *Tg(hsp70l:tBFP-2a-nkx3.1)*$^{ca91}$, *Tg(UAS:cxcl12a)*$^{ca504}$. First, middle entry clones of tagBFP (Evrogen), tagBFP PCR-fused to *nkx3.1* or *cxcl12b* were made by amplification using primers in S4 Table and cloned into pDONR221 using BP Clonase (Thermo Fisher). Transposon Tol2 vectors were assembled using LR Clonase (Thermo Fisher) and the Tol2 vector [53]. *Tg(tbx6:cre;myl7:GFP)*$^{ca92}$ and *Tg(sox10:cre;myl7:GFP)*$^{ca93}$ were injected in our laboratory from Tol2 plasmids provided by Tom Carney [54]. *Tg(*hsp70l-loxP-STOP-loxP-H2B-GFP_cryaa-cerulean)$^{ca94}$ was created from Addgene plasmid 24334 [55] by digestion with XhoI and religation. This deleted mCherry but maintains the stop sequences and was followed by injection into zebrafish and raising of F1 and further generations from the F0 founder.

The *nkx3.1*$^{ca116}$ mutant was generated using CRISPR/Cas9 as previously described [56]. Target sites were identified using CHOPCHOP ([57]). The guide sequence was 5′-GGGGAGGCGGGAAAAAGAAGCGG -3′. To assemble DNA templates for sgRNA transcription, gene-specific oligonucleotides containing the T7 promoter sequence (5′-TAATAC-GACTCACTATA-3′), the 20-base target site, and a complementary sequence were annealed to a constant oligonucleotide encoding the reverse-complement of the tracrRNA tail. sgRNAs were generated by in vitro transcription using the Megascript kit (Ambion). Cas9 mRNA was transcribed from linearized pCS2-Cas9 plasmid using the mMachine SP6 kit (Ambion). To generate mutants, one-cell stage wild-type embryos were injected with a mix containing 20 ng/μl sgRNA and 200 ng/μl Cas9 mRNA. Injected fish were raised to adulthood and crossed to generate F1 embryos. T7 Endonuclease I assay (NEB) was then used to identify the presence of indel mutations in the targeted region of F1 fish. *nkx3.1*$^{ca116}$ has a 13-bp deletion (S2 Fig).

### Microscopy and imaging analysis

Embryos were sampled randomly from a clutch and included hemorrhaged and nonhemorrhaged embryos. Embryos were imaged with a 20× (NA 0.8) objective on an inverted Zeiss LSM880 Airyscan or LSM900 laser confocal microscope mounted in 0.8% low-melt agarose on glass-bottomed Petri dishes (MatTek). The imaging field was adjusted and tiled to encompass the entire brain vasculature (rostral-caudal and dorsal ventral) as marked with *Tg(kdrl:mCherry)*). For analysis, images were assembled using ImageJ [58]. Briefly to identify pericytes without overlap, the stack was divided into 2 dorsal and ventral substacks that cover the entire depth of mid- and hindbrain CtAs (central arteries). Pericytes (defined as transgene-labelled cells with a soma and processes) were counted manually on these stacks ensuring that no pericyte was counted twice. The numbers were compiled to obtain the total number of pericytes. Blood vessel network length was calculated automatically using VesselMetrics, a Python program [59]. All computational analysis was accompanied by manual inspection of the data,

removing any erroneous segments (unlumenized angiogenic sprouts). Pericyte density was obtained by dividing the number of pericytes over the vessel network length.

### Statistical analysis

Statistical analysis used GraphPad Prism 7 software, using a one-way ANOVA with multiple comparisons and a Tukey's or Dunnett's post hoc test. Results are expressed as mean ± SD. The N's of biological and n's of technical replicates, *p*-values, and statistical test used are provided in the figure legends. All raw data underlying figures can be found in S3 Table.

### Lineage analysis

Stable transgenic reporter lines harboring *Tg(*hsp70l-loxP-STOP-loxP-H2B-GFP_cryaa-cerulean)[ca94] (nuclear, GFP), (TgBAC(pdgfrb:Gal4FF) or TgBAC(nkx3.1:Gal4)) and Tg(UAS:ntr: mCherry) (cytoplasmic, pericyte, red) were crossed with *Tg(tbx6:cre;myl7:GFP)[ca92]* or *Tg (sox10:cre;myl7:GFP)[ca93]*. Fish were imaged at 75 hpf and double positive pericytes counted separately in midbrain and hindbrain.

### In situ hybridization

For hybridization chain reaction (HCR) in situ hybridization, custom probes for *nkx3.1*, *tbx18*, *foxf2a*, *cxcr4a*, and *cxcl12b* were obtained from Molecular Instruments (Los Angeles, CA). The in situ staining reactions occurred as recommended by the manufacturer.

### Drug treatments

All small molecules are listed in S5 Table. AMD 3100 was dissolved in E3 zebrafish water (5 mM NaCl, 0.17 mM KCl, 0.33 mM $CaCl_2$, 0.33 mM $MgSO_4$, 0.00001% Methylene Blue) and applied at 100 μM from 24 to 75 hpf in the dark. Metronidazole was dissolved at 5 mM in E3 and was applied from 24 to 48 hpf in the dark. Embryos were washed out and grown in E3 medium without the drugs until imaging.

### Embryo dissociation and FACs sorting

Two biologically independent replicates were dissociated, FACs sorted, library prepped, and sequenced independently. Each replicate started with approximately 250 wild-type 30 hpf zebrafish embryos on the TL background expressing the *Tg(nkx3.1:Gal4;UAS:NTR-mCherry)[ca101]* transgene [22]. Embryos were anaesthetized in Tricaine, kept on ice, and then dissociated with 0.25% Trypsin in 1 mM EDTA at 28°C with 300 rpm shaking for 20 minutes. The reaction was stopped using 1.25 μL 0.8M $CaCl_2$ and 100 μL 1% FBS. Cells were washed in 1% FBS in Dulbecco's PBS 3 times before straining through 75 μm and 30 μm strainers (Greiner Bio-One and Miltenyi Biotec) in Dulbecco's PBS. Cell viability was determined using Trypan blue (Sigma) and was greater than 80% for both replicates. Cells were subjected to FACS (BD Facs Aria III, BD Biosciences) sorting for mCherry and excluding doublets and nonviable cells.

### scRNA-Seq library construction, sequencing

The biologically independent replicates of FACS-sorted nkx3.1-positive cells were each made into an independent library using the Chromium Single Cell Chip A kit, the Chromium Single cell 3′ Library & Gel beaded kit V3 and 3.1 Next GEM (10× Genomics; Pleasanton, CA, USA). Libraries were sequenced on the Illumina NovaSeq using an S2 Flowcell (Illumina) at the Centre for Health Genomics and Informatics, University of Calgary. A total of 3,359 cells passed quality control for analysis, 2,129 cells from sample 1 and 1,230 from sample 2. The data from

both samples were integrated using Cellranger_aggr. All raw FASTQs were aligned to the zebrafish reference genome GRCz11 (version 4.3.2) generated using the Cellranger version 3.1.0 mkref pipeline. The gene-barcode matrix output was processed using a standard Seurat v3 pipeline in R [60]. Quality control steps filtered out cells expressing fewer than 200 genes or more than 2,500 genes, as well as cells that had a mitochondrial content <5%. UMAP with 15 principal components and resolution of 0.6 was used for dimension reduction. To assess the reproducibility of the 2 samples, cells from each replicate were projected onto the UMAP and proportions of cells in each cluster compared between samples (S6 Fig). The R code for this analysis is in the S1 Data.

### Cluster assignment

Clusters were manually assigned by querying top markers in each cluster with known markers from annotated datasets [28–31,61]. A list of the cluster number, assignment, genes used for assignment, and references for known cluster markers can be found in S1 Table.

### RNA velocity analysis

The RNA velocity pipeline for Seurat objects (Satija lab) was implemented. Loom files were generated from position-sorted aligned BAM files using velocyto (version- 0.17.15) [62]. A combined loom file containing counts of spliced, unspliced, and ambiguous transcripts of both samples was read using Seurat function ReadVelocity. A count table of spliced and unspliced transcripts (stored as a Seurat object) was processed to merge with the existing Seurat object generated for cluster identification. Velocity analysis was performed on the integrated Seurat object using velocyto.R [62]. RNA velocity vectors were projected on UMAP embeddings generated during cluster generation. The python code for this analysis is in S2 Data.

### Bulk RNA sequencing

Approximately 30 hpf MZ $nkx3.1^{ca116}$ mutant and wild-type embryos were collected and RNA prepared using Trizol (Thermo Fisher, Waltham MA). Libraries were prepared from 50 ng/μg/μl of total mRNA per sample using the Illumina Ultra II directional RNA library prep (San Diego, CA) and sequenced via the NovaSeq SP 100 cycle v 1.5 sequencing run with 33 million reads per sample. Four replicates were sequenced per condition.

For bulk RNASeq, quality control evaluations performed by FastQC v0.11.9 [63] revealed that the reads were of high quality and required no trimming step. Consequently, the reads were mapped to the reference genome danRer11 obtained from UCSC and gene annotation file Zebrafish ENSEMBL Lawson v4.3.2 [64] using the splice-aware alignment tool, STAR v2.7.8a [65]. STAR was run on default settings with the additional optional command '—quantMode GeneCounts' to generate the gene counts files. The gene counts files were filtered for genes with less than 10 counts and normalized to the median ratio using DESeq2 v1.34.0 [66]. Differentially expressed genes were identified as genes with FDR-adjusted $p$-value < 0.05 and log2(fold-change) < −0.5 or log2(foldchange) > 0.5. Volcano plots of differentially expressed genes was generated using Enhanced Volcano v1.12.0 [67].

## Supporting information

**S1 Movie. Time-lapse of labelled Tg(*nkx3.1^{NTR-}mcherry*) perivascular cells as they migrate and proliferate on blood vessels from 55–65 hpf.**
(AVI)

**S1 Table. 30 hpf scRNAseq (see Excel sheet).**
(XLSX)

**S2 Table. 30 hpf bulk RNAseq (see Excel sheet).**
(XLSX)

**S3 Table. Raw data (see Excel spreadsheet).**
(XLSX)

**S4 Table. Primers, guides, and HCR probes.**
(XLSX)

**S5 Table. Reagent table.**
(XLSX)

**S1 Fig. Lineage tracing of *nkx3.1*-expressing cells and pericytes at 75 hpf. (A**, **D**, **G**, **J**). Schematics of lineage tracing strategy showing how Tg(tbx6:Cre) or Tg(sox10:Cre) drivers crossed to the Tg(loxp-stop-loxp-H2B-GFP) reporter labels progeny with nuclear GFP. Pericytes TgBAC(pdgfrb:Gal4FF) or nkx3.1-expressing cells TgBAC(nkx3.1:Gal4) +Tg(UAS:ntr: mCherry) label cytoplasm red. (**B**-**L**) All images are dorsal views of embryonic mid or hind brains at 75 hpf, as marked. (**B**, **C**, **E**, **F**) Mesodermal lineage trace of pericytes (**B**, **C**) and *nkx3.1* cells (**E**, **F**). (**H**, **I**, **K**, **L**) Neural crest lineage trace of pericytes (**H**, **I**) and *nkx3.1* cells (**K**, **L**). Arrowheads indicate double positive *pdgfrβ* or *nkx3.1* cells. Scale bar is 50 μm.
(PDF)

**S2 Fig. *nkx3.1*ca116 mutation characterization.**
(PDF)

**S3 Fig. Zygotic *nkx3.1* mutants have wild-type pericyte numbers at 75 hpf. (A**, **B**) Dorsal views of embryonic brain of zygotic *nkx3.1* mutants showing no change in brain pericyte numbers as compared to wild type. Pericytes (green, arrowheads) are labelled with TgBAC(pdgfrβ: GFP) and vessels (red) are labelled with Tg(kdrl:mCherry). (**C**) Quantitation of pericyte numbers shows no significance using one-way ANOVA with Tukey's test. ($n$ = 8 wild types, 17 heterozygotes, and 10 mutants). Scale bar is 50 μm. The data underlying this figure can be found in S3 Table.
(PDF)

**S4 Fig. Central artery vessel network length is unchanged across multiple experimental manipulations.** Vessel network length of the hindbrain CtAs was measured using VesselMetrics. Genotypes and treatments are labelled. No treatment or mutant significantly alters the endothelial vessel length. Statistics used a Student *t* test. The data underlying this figure can be found in S3 Table.
(PDF)

**S5 Fig. Pericyte number and density is decreased at 5dpf in *nkx3.1* maternal-zygotic mutants. (A**, **B**) Dorsal views of embryonic brain of MZ *nkx3.1* mutants at 5 dpf. Pericytes (green, arrowheads) are labelled with TgBAC(pdgfrβ:GFP) and vessels (red) are labelled with Tg(kdrl:mCherry). There are significantly fewer brain pericytes (**C**) and reduced pericyte density (**D**) in nkx3.1−/− mutant as compared to nkx3.1+/− controls. Statistics used a Student *t* test. ($n$ = 9 wild types, 6 mutants). Scale bar is 50 μm. The data underlying this figure can be found in S3 Table.
(PDF)

**S6 Fig. Analysis of proportions and batch effects analysis of scRNAseq.** (**A**) Overlay of UMAP projections from 2 biologically independent scRNAseq samples (1, red; 2, blue) showing cells with similar distributions from both samples. (**B**) Stacked barplot showing the proportion of cells in each cluster deriving from the first sample [1] or second sample [2] as normalized as a percentage to the total number of cells from each sample. (Right) Table of proportions of the different cell types in each sample.
(PDF)

**S7 Fig. Dotplot showing expression of common fibroblast markers in Progenitor-2 and Fb clusters.**
(PDF)

**S8 Fig. Featureplots of genes enriched in the Fb-V scRNAseq cluster (fibroblast-like pericyte precursors).**
(PDF)

**S9 Fig. Featureplots of genes enriched in the Fb-A scRNAseq cluster.**
(PDF)

**S10 Fig. Featureplots of genes enriched in the Fb-B scRNAseq cluster.**
(PDF)

**S11 Fig. Featureplots of canonical pericyte markers in the nkx3.1-positive cell scRNAseq data.**
(PDF)

**S12 Fig. Expression analysis of *nkx3.1* and *cxcl12b* using HCR.** (**A**) Dorsal view of the posterior head region of 16 hpf embryo showing expression overlap (white bracket) between *cxcl12b* (green) and *nkx3.1* (red). (**B**) Lateral view of the posterior head region of 24 hpf embryo showing expression overlap (white bracket) between cxcl12b and nkx3.1. A-Anterior, P-Posterior, D-Dorsal, V-Ventral. Scale bar is 50 μm.
(PDF)

**S13 Fig. cxcl12b mRNA injection increases pericyte numbers at 75 hpf.** (**A**-**C**) Dorsal views of embryonic brain of uninjected control (**A**) and *cxcl12b* mRNA injected (**B**) embryos showing an increase in brain pericyte number. (**C**) Quantitation of brain pericytes (*n* = 5 wild-type and 9 injected embryos). (**D**-**F**) Dorsal views of embryonic brain of uninjected *nkx3.1*−/− (**D**) and *cxcl12b* mRNA injected nkx3.1−/− embryos (**E**) showing an increase in brain pericyte number due to *cxcl12b* mRNA injection (**F**, *n* = 10 uninjected and 10 injected mutants). Pericytes (green, arrowheads) are labelled with TgBAC(pdgfrβ:GFP) and vessels (red) are labelled with Tg(kdrl:mCherry). (**G**) Hemorrhage rates were unchanged in mutants at 48 hpf (*N* = 3 with 141 uninjected at 167 injected mutants analyzed). Statistics used the Student *t* test. Scale bar is 50 μm. The data underlying this figure can be found in S3 Table.
(PDF)

**S14 Fig. No overlap between *cxcl12b* and *cxcr4a* mRNA at 36 hpf.** Dorsal view of the ventral head region of 36 hpf embryo showing no expression overlap between *cxcl12b* (green) and *cxcr4a* (red). A-Anterior, P-Posterior, Scale bar is 20 μm.
(PDF)

**S15 Fig. Expression of pericyte precursor genes in the Daniocell database.** Output of searches for early pericyte markers showing the pericyte cluster expression at 24 hpf–48 hpf of

the indicated genes (red box).
(PDF)

**S1 Data. R Script for analysis of scRNAseq data.**
(DOCX)

**S2 Data. Python Script for velocity analysis.**
(DOCX)

## Author Contributions

**Conceptualization:** Suchit Ahuja, Sarah J. Childs.

**Funding acquisition:** Jeff Biernaskie, Peng Huang, Sarah J. Childs.

**Investigation:** Suchit Ahuja, Cynthia Adjekukor, Katrinka M. Kocha.

**Methodology:** Suchit Ahuja, Cynthia Adjekukor, Qing Li, Katrinka M. Kocha, Nicole Rosin, Elodie Labit, Sarthak Sinha, Ankita Narang.

**Project administration:** Sarah J. Childs.

**Resources:** Sarah J. Childs.

**Supervision:** Quan Long, Jeff Biernaskie, Peng Huang, Sarah J. Childs.

**Writing – original draft:** Suchit Ahuja, Sarah J. Childs.

**Writing – review & editing:** Suchit Ahuja, Cynthia Adjekukor, Jeff Biernaskie, Peng Huang, Sarah J. Childs.

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
