## [Editor Report · Decision Letter 0]

27 Jun 2023

Dear Dr Childs, 

Thank you for submitting your manuscript entitled "A genetic program for vascular pericyte precursors" for consideration as a Research Article by PLOS Biology.

Your manuscript has now been evaluated by the PLOS Biology editorial staff as well as by an academic editor with relevant expertise and I am writing to let you know that we would like to send your submission out for external peer review.

**In addition to providing the metadata related to your submission, we also have an editorial suggestion, which we hope you will consider before review. After discussion with the Academic Editor, we think that it would be valuable for you to move some of the data showing the functional consequences of your manipulations in the main figures. For example, we think the data showing that several manipulations cause brain hemorrhage should be added to the main figures. 

If you agree, please make these changes as you complete your submission. I have extended the deadline for you to complete the submission of your manuscript by a week (due by Jul 05 2023 11:59PM). If you anticipate that these changes will take much longer than that, please do get in touch. Due to a quirk in our system, if the revision before review would require much longer than a week, we may need you to withdraw the submission and resubmit when the revisions are done. We would then send the revised manuscript out for review. 

To provide the metadata for your submission, please Login to Editorial Manager (https://www.editorialmanager.com/pbiology). Once your full submission is complete, this metadata will undergo a series of checks in preparation for peer review. After your manuscript has passed the checks it will be sent out for review.

As another note, if your manuscript has been previously peer-reviewed at another journal, PLOS Biology is willing to work with those reviews in order to avoid re-starting the process. Submission of the previous reviews is entirely optional and our ability to use them effectively will depend on the willingness of the previous journal to confirm the content of the reports and share the reviewer identities. Please note that we reserve the right to invite additional reviewers if we consider that additional/independent reviewers are needed, although we aim to avoid this as far as possible. In our experience, working with previous reviews does save time. 

Kind regards,

Luke

Lucas Smith, Ph.D.

Senior Editor

PLOS Biology

lsmith@plos.org

---

## [Decision Letter · Decision Letter 1]

21 Aug 2023

Dear Dr Childs,

Thank you for your patience while your manuscript "A genetic program for vascular pericyte precursors" was peer-reviewed at PLOS Biology. Your manuscript has been evaluated by the PLOS Biology editors, an Academic Editor with relevant expertise, and by several independent reviewers.

As you will see in the reviewer reports, which can be found at the end of this email, although the reviewers find the work potentially interesting, they have also raised a number of important concerns. Based on their specific comments and following discussion with the Academic Editor, it is clear that a substantial amount of work would be required to meet the criteria for publication in PLOS Biology and to strengthen the conclusions of the study. However, given our and the reviewer interest in your study, we would be open to inviting a comprehensive revision of the study that thoroughly addresses the reviewers' comments.

After discussion with the Academic Editor, we think that it will be important to experimentally strengthen the conclusions in response to reviewer concerns, by providing more in-depth quantifications of the vessel patterning and quantifying pericyte density. We also think that you should provide data from additional time points - although the Academic Editor has commented that perhaps not everything needs to be done at each time point, as long as data is provided to test critical conclusions at these timepoints.

Regarding reviewer 3's suggestion that nkx3.1 be linked to Notch 3 expression - we think that this is an interesting line of research and would encourage the experiment if it is easily doable. However, the Academic Editor has also commented that it would also be OK to address this point through discussion if it is not.

As a last point, we would also like to suggest again, the idea of moving the hemorrhaging data to the main figures, as we think this will be of interest to a general audience. This would also fit with the theme of providing more characterization of the animals as suggested by the reviewers.

Given the extent of revision that would be needed, we cannot make a decision about publication until we have seen the revised manuscript and your response to the reviewers' comments. Your revised manuscript would need to be seen by the reviewers again, but please note that we would not engage them unless their main concerns have been addressed.

We appreciate that these requests represent a great deal of extra work, and we are willing to relax our standard revision time to allow you 6 months to revise your study. Please email us (plosbiology@plos.org) if you have any questions or concerns, or envision needing a (short) extension.

**IMPORTANT - SUBMITTING YOUR REVISION**

*Resubmission Checklist*

*Published Peer Review*

*PLOS Data Policy*

*Blot and Gel Data Policy*

Sincerely,

Lucas

Lucas Smith, Ph.D.

Senior Editor

PLOS Biology

lsmith@plos.org

REVIEWS:

Reviewer #1: Summary:

In this manuscript, Ahuja et al. have sought to define a precursor cell population that differentiates into brain pericytes and identify genetic programs responsible for this process. This is an outstanding question in cerebrovascular biology, thus this study provides novel and important biological insights in this field. In this manuscript, the authors specifically focused on investigating an early embryonic cell population that expresses the transcription factor Nkx3.1 and determining how this cell lineage differentiates into brain pericytes. This study has been carried out by combining novel molecular genetic zebrafish tools with bulk and single-cell transcriptomic analyses through LOF, GOF, global or cell type-specific rescue experiments, and cell lineage tracing. A variety of the modern and elegant techniques employed in this manuscript are impressive and make this study stand out.

Overall, this study is very well-crafted, nicely presented, and well described. Experiments were carefully designed and executed, making the findings of this manuscript solid and rigorous in most parts. Moreover, this study seeks to fill an important gap in our scientific knowledge by revealing the developmental trajectories of brain pericytes. This is a crucial next step in both developmental neurobiology and vascular biology fields, and their findings will provide valuable insights into brain pericyte development. Despite the significance of this study, however, I have several critical points that will need to be addressed by the authors to improve the current format of the manuscript.

Major Comments:

1) Most of the major conclusions the authors have drawn in this manuscript are based on their quantifications for the number of brain pericytes, which were conducted at the single time point 75 hpf. Since the authors showed disrupted patterning of brain vascular networks in some, or most, loss- and gain-of-function genetic experiments and after pharmacological treatments, the reduced number of brain pericytes could be due to the mis-patterning and/or reduced complexity of brain vascular networks. Thus, it would be more appropriate to present these quantification results by pericyte density per vessel length rather than absolute pericyte numbers that may rely on the extent of vascular network elaboration. Presenting both quantifications will be more informative and strengthen the findings.

2) Related to my major point #1, phenotypic assessments at a single-time point raises a concern about data interpretation of observed phenotypes. For example, a MZ nkx3.1 mutant embryo at 52 hpf (Fig. 2B) displays peri-cardiac edema, bigger yolk sac, and brain ventricular enlargement (hydrocephalus) compared to the heterozygous animal (Fig. 2A), raising a concern that there is a significant developmental delay in the mutant partly due to impaired blood circulation resulting from peri-cardiac edema and weakened heartbeats. Similarly, 75 hpf larvae (Fig. 2D and 2E) showed a substantial difference in their brain size (much smaller brain in the mutant). Again, this brain size difference raises a concern about the interpretation of the authors' quantification data because significantly mismatched developmental stages were potentially used for quantifications and phenotypic comparisons. This is a critical point because pericyte reduction in MZ nkx3.1 mutants is much milder than the previously reported phenotypes observed in pdgfrb and notch3 zebrafish mutant larvae, which exhibit a near-complete or severe loss of brain pericytes (Ando et al., Dev. Biol., 2021; Wang et al., Development, 2014). The bottom line is that phenotypic analysis conducted at only a single time point makes the conclusions weak in some contexts due to the concerns I raised above. The authors are recommended to analyze phenotypes at additional time point(s) to address this type of concerns, especially for brain pericyte number quantifications (Fig. 2F). 

3) Related to the authors' proposed model, if Nkx3.1 acts upstream of Cxcl12-Cxcr4 signaling that may subsequently regulate the Pdgfrb signaling pathway, how do the authors explain the much milder pericyte-loss phenotypes observed in MZ nkx3.1 mutants and after AMD3100 treatment (Cxcr4 inhibition) compared to pdgfrb mutants? This point and apparent discrepancy should be discussed. 

4) Loss of cxcl12b expression in nkx3.1 mutant embryos is drastic and convincing (Fig. 4A, 4B), but it is not clear how this secreted ligand controls pericyte differentiation process in Nkx3.1-positive precursors. Given the crucial role of this chemokine signaling in vSMC recruitment in the dorsal aorta of the zebrafish trunk (Stratman et al., Commun. Biol., 2020) combined with endothelial cxcl12b expression shown in Fig. 3N, it is possible that endothelial cells-derived Cxcl12b contributes to brain pericyte recruitment. Although rescue experiments are elegantly performed in Fig. 5, this part of the study is still under-developed partly due to the limited spatial information on cxcl12b and cxcr4 expression and their cellular sources. It would be much more informative if the authors would perform HCR co-expression analysis of cxcl12b and cxcr4a, and/or their individual expression analysis in a vascular endothelial reporter background and/or together with a nkx3.1 probe. These data will likely strengthen this part of the study and provide data to suggest a better cellular and signaling model.

5) Are the images that showed expression overlaps of two genes (Fig. 3E-M and Fig. S10) presented as z-stack projection or single-plane images? Expression overlap should be shown on a single-plane frame. Please also clarify this information in the legend.

6) No descriptions of the quantification means are provided in the Methods section. Please provide detailed descriptions on how all the quantifications presented in this manuscript were performed.

Minor Comments:

1) Please provide explanations on the specificity of TgBAC(nkx3.1:Gal4)ca101 expression patterns in the developing brain as the cited original paper did not focus on the brain.

2) In Fig. 1A-C panels, please add the indication of A-P and D-V axes, similar to Fig. S10.

3) In Fig. 2A-C legends, please include animal numbers for the nkx3.1+/- genotype. 

4) In Fig. 5F graph, please add the P-value information.

Reviewer #2: General comments:

The authors identified the transcription factor Nkx3.1 as regulating development of mature pericytes from precursors. By lineage tracing experiments, they identified that nkx3.1 is expressed in precursor cell populations which later developed into pericytes. By conducting a rescue experiment, the authors showed that Cxcl12b acts downstream of Nk3.1. Overall, the study used a combination of genetics and transcriptomics and proposed a mechanism where early pericyte differentiation is regulated by Nk3.1-cxcl12b signalling.

While several molecular pathways are known to be involved in pericyte development (pdgfrb, notch) there is a lack of understanding of early transcriptional regulators and differentiation. This study would identify an important transcription factor in pericyte development and at least one key target in cxcl12b. This finding is likely to be well received by the vascular biology field and open up new understanding in pericyte development. 

In my opinion, this is an extensive study and presents novel findings in pericyte development. However, there are still several issues with data and interpretation. See below. 

Major:

1. The authors mention that nkx3.1 MZ mutants display mispatterned vessels. This is a vague term that needs to be addressed. Furthermore, there is no quantification and the representative images do not show an obvious defect. The authors need to carefully examine and quantify the vascular phenotypes such as volume, coverage and branching. Furthermore, all pericyte quantifications need to be adjusted to be expresses relative to vascular coverage in order to rule out the possibility that changes in pericyte numbers are due to altered vasculature overall. 

2. Does nkx3.1 generally impact all early mesenchymal fibroblasts development? Or just the precursors of the pericytes? Quantification of pdgfrB low expressing cells in the mutants or using other fibroblast markers would give more confidence in the specificity of the phenotype to pericytes. 

3. The authors should give further details about the gross morphological defects displayed in MZ nkx3.1 mutants. Do these animals survive the larval stages? Do they have delayed development? Is there a restricted blood supply to the brain or other potentially affected tissues that in turn cause a general reduction in tissue size, vessel formation and pericyte development? 

4. The authors claim that hemorrhage in nkx3.1 MZ mutants occurs as a result of pericyte loss. Ando et al 2022 showed that pdgfrb mutants, which lack most brain pericytes except those on the metencephalic artery, do not display hemorrhage until juvenile stages. Considering that pericyte loss in pdgfrb mutants is more severe than in nkx3.1 MZ mutants, the authors claim seems very unlikely to be correct. The nature and cause of the haemorhages should be characterised more carefully. Do the

---

## [Decision Letter · Decision Letter 2]

12 Feb 2024

Dear Dr Childs,

Thank you for your patience while we considered your revised manuscript "A genetic program for vascular pericyte precursors" for consideration as a Research Article at PLOS Biology. Your revised study has now been evaluated by the PLOS Biology editors, the Academic Editor and the original reviewers. 

As you will see in their comments, which are appended below, the reviewers agree that the study has been strengthened in the last round of revision but they have also identified a number of lingering issues. We do not think that the last concerns will require additional analyses or experiments, but we do think they are important and will need to be carefully and thoroughly addressed before we can consider your study for publication. 

Therefore, in light of the reviews we are pleased to offer you the opportunity to address the remaining points from the reviewers in a revision that we anticipate should not take you very long. We will then assess your revised manuscript and your response to the reviewers' comments with our Academic Editor aiming to avoid further rounds of peer-review, although might need to consult with the reviewers, depending on the nature of the revisions.

**IMPORTANT: As you address the last reviewer requests, we also ask that you attend to the following editorial requests: 

1) TITLE: We think the title of your piece would be strengthened by including more details of your findings. Therefore, if you agree, we suggest you change it to something like: 

"The development of brain pericytes requires expression of the transcription factor nkx3.1 in intermediate precursors"

2) ABSTRACT: Please note that per journal policy, the model system/species studied should be clearly stated in the abstract of your manuscript

3) FINANCIAL DISCLOSURES: Please update the financial disclosures statement, in our online system, to describe the role of any sponsors or funders in the study design, data collection and analysis, decision to publish, or preparation of the manuscript. If the funders had no role in any of the above, include this sentence at the end of your statement: "The funders had no role in study design, data collection and analysis, decision to publish, or preparation of the manuscript.

4) DATA: Thank you for providing the underlying data as a deposition to GEO and as Supplemental Table 3. These data meet our reporting requirements, but I have a few minor requests: 

 - I see the GEO datasets are set to private. These will need to be made publicly available upon publication (which I suspect you were planning to do, but just flagging this as a reminder)

 - Can you please add a sentence to each relevant figure legend, pointing readers to where the underlying data can be found? For example, you can add the sentence "the data underlying this figure can be found in Supp Table 3"

 - I noticed that in your methods, you reference a S5 Table as containing the raw data for your paper. I suspect this is a type, and maybe you meant to reference Table S3? 

5) CODE: Per journal policy, if any code was generated for this study to support the conclusions of your manuscript, we would require that you make it available without restrictions upon publication. Please ensure that any code is sufficiently well documented and reusable, and that your Data Statement in the Editorial Manager submission system accurately describes where your code can be found.

**IMPORTANT - SUBMITTING YOUR REVISION**

*Resubmission Checklist*

*Published Peer Review*

*Blot and Gel Data Policy*

Sincerely,

Lucas

Lucas Smith, Ph.D.

Senior Editor

PLOS Biology

lsmith@plos.org

REVIEWS:

Reviewer #1: The authors have satisfactorily addressed all the concerns I originally raised. 

However, I noticed that the Method section still lacks the descriptions of several important procedures, including the AMD3100 pharmacological treatment experiments (related to Fig. 4D-G) and the ablation experiments of nkx3.1-positive cells using the NTR-MTZ cell ablation method (related to Fig. 2M-Q).

The authors should include the detailed information on these missing experimental procedures in the Method section.

Reviewer #2: The authors made substantial improvements that have addressed many of the reviewer comments and have provided further data on the role of Nkx3.1-cxcl12b signalling in pericyte development. 

My comments have been mostly addressed, however there are some minor issues:

* Responses related to queries 2-2 and 2-5: The authors showed in the paper that Nkx3.1-cxcl12b signalling regulates brain pericyte development, however it is clear from their response that the phenotypes are broader and not exclusively specific to pericytes. The authors should clearly discuss this in the manuscript. Nkx3.1's broader role in fibroblasts and in different tissues need to be emphasized as well.

* Related to query 2-8: To definitively interpret a rescue experiment, heterozygotes need to be present in the same experiment. This is needed to show that the mRNA rescue returns LOF phenotype to the equivalent of normal levels. Without this control, the definition may be limited to "an increase in the injected group compared to the nkx3.1-/- mutants" rather than a rescue. 

* FigS5: Fig legend defines animals as maternal-zygotic in the bold section, however in the further description says they are zygotic. Which is it? Also, C is missing stats on the graph. 

* Some minor points such as typos and stats need to be addressed by going through the manuscript and the figures thoroughly. Some examples are

o Fig2C hemorrhage spelling, inconsistent font style and size, nkx3.1 spelling in the legend.

o Last paragraph in the "Nkx3.1 function is required for brain pericyte development" was unclear due to potentially misplaced edits and incomplete sentences.

Reviewer #3: The authors have addressed some of the concerns raised in the previous review. However, the manuscript could still benefit from further clarification of experimental details and improved writing clarity. I apologize that some of my comments address points I missed in the first revised version, which I should have identified earlier.

I could not locate the Supplementary tables referenced in the manuscript, so I was unable to review these tables.

1. Abstract: Please consider rephrasing the first sentence in Abstract. The wording "regulating" implies influencing, not guaranteeing. Pericytes may regulate endothelial function in a way that promotes efficient blood flow under normal conditions but they are not the only players. Blood flow regulation is a complex process affected by multiple factors. So, even if pericytes function optimally, other issues could still impair blood flow.

2. Page 5 "Pericytes in both hindbrain and midbrain express the nkx3.1 transgene (Fig. 1D)." 

I find it difficult to identify nkx3.1-expressing cells in the midbrain region of your image. While you provided an enlarged image of the hindbrain region demonstrating nkx3.1 progeny, I would appreciate (and likely also other readers) a similarly enlarged image of the midbrain for clearer observation.

Additionally, your statement on page 5 regarding nkx3.1 expression by pericytes in the hindbrain /" Pericytes in both hindbrain and midbrain express the nkx3.1 transgene (Fig. 1D)"/ - The image in Fig. 1D depicts a 4 dpf stage, where nkx3.1 expression is no longer present. However, mCherry is present beyond nkx3.1 expression, and serves as an indicator of cells that previously expressed nkx3.1. Therefore, to accurately describe the observed data, the sentence should be rephrased accordingly.

3. S Fig1 - Please consider labelling "Midbrain" and "Hindbrain" and not "Mid Brain" and "Hind Brain"

4. Fig. 2C-Please correct the misspelling - "hemmorage to "Hemorrhage".

5. In your response to a reviewer's comment, you mentioned a reduced lifespan for Nkx3.1-/- fish (1 year). Including this information directly in the main text (on page 6) would enhance clarity and provide valuable context. If possible, including quantitative data would further strengthen your claim.

6. Figure legend 2: "(E-F) Dorsal images of wildtype and nkx3.1 mutant embryo expressing Tg(pdgfrβ:GFP) and Tg(kdrl:mCherry) showing fewer brain pericytes (arrows) in mutants at 75 hpf."

Please clarify which image (F or E) shows the mutant and which one wildtype?

Additionally, please specify whether you used wild-type or heterozygous embryos as controls in your experiment.

7. Figure 2FE, IJ and 4DE. I struggle to see differences between the pericyte numbers in the provided images. All images (2F vs. 2E and 2I v.s 2J and 4D vs. 4E) have an equal number of arrowheads. Despite all panels pointing to the same number of pericytes (four arrowheads), the authors claim panels 2E, 4E have fewer and panel 2J have more pericytes. This presentation format hinders clarity. I am unable to see these differences in the current figures. Please consider revising the data visualization for improved comprehension.

8. Supplementary Fig. 5 "There are significantly decreased brain pericyte numbers (C) and pericyte density (D) as compared to wildtype". 

The figure legend states comparison to wildtype data, but only heterozygote and knockout data are shown. Please include wildtype data or revise the legend to reflect the available data.

9. Scheme in Figure 3. - The figure legend lacks an explanation of what the schematic represents. Additionally, the meaning of the grey ovals is unclear. Please provide clarification in the legend.

10. Response to reviewers, point 3.1.

While you mentioned this is the standard way of presenting data, I believe reaching an audience beyond zebrafish experts is valuable. Including all details about how the experiments were performed and data analysed would benefit them considerably. Additionally, providing these details in the Materials and Methods section aligns with transparency best practices and allows readers to fully understand the experimental procedures.

"Supplementary Figure 12 (G) Hemorrhage rates were unchanged in mutants at 48 hpf (N=3 with 141 uninjected at 167 injected mutants analyzed)" - Please report the individual embryo counts used in each experiment, rather than only providing the total number.

This raises another point: how were individual embryos selected for pericyte quantification? Did you specifically analyze those with hemorrhages, or exclude them? While the field might have standard practices like excluding hemorrhagic embryos, your Methods section lacks clarity on this aspect.

11. Response to reviewers, point 3.4.

To ensure clarity, I recommend providing more detail in the Materials and Methods section regarding cell sorting for sequencing. Specifically, it is unclear whether cells were sorted in two separate rounds. Given that you generated two independent libraries, I assume this was the case. If so, please clarify the number of embryos used per preparation (250 per run?)

Please show how cells from the two different batches (libraries) are distributed on the UMAP and give the percentage of cells from each batch in corresponding clusters.

I am also following up on my previous request to provide a more detailed explanation of the cluster annotation. Unfortunately, my previous comment about this has not been addressed. For example, "63% of the cells form a group of connected clusters that appear to be progenitors (Prog-1/2) and more differentiated cells" - to ensure clarity for readers unfamiliar with the specific field, I suggest providing additional details regarding the cluster annotations. Explaining the reasoning would greatly help readers understand your data and interpretation.

12. Mouse gene name should be written "Tbx18" (italics) and not "tbx18" (italics) (page 8).

13. Response to reviewers, point 3.9

Please provide a detailed description of the image acquisition for pericyte quantification. Include the numerical aperture of the objective.

Please explain how you identified pericytes in the images. Please specify the precise region or area where pericytes were quantified.

While citing reference 57 for the detailed method of image analysis, please provide a brief summary of how pericyte quantification (number and density) was performed.

Could you clarify what constitutes "erroneous segments"?

The pericyte density axis label needs clarification. What does the numerical value represent? Specify what unit this represents (e.g., pericytes per area unit, etc.).

---

## [Editor Report · Decision Letter 3]

14 Mar 2024

Dear Dr Childs,

Thank you for the submission of your revised Research Article "The development of brain pericytes requires expression of the transcription factor nkx3.1 in intermediate precursors" for publication in PLOS Biology and thank you for addressing the last reviewer and editorial requests in this revision. On behalf of my colleagues and the Academic Editor, Cody J. Smith, I am pleased to say that we can in principle accept your manuscript for publication, provided you address any remaining formatting and reporting issues. These will be detailed in an email you should receive within 2-3 business days from our colleagues in the journal operations team; no action is required from you until then. Please note that we will not be able to formally accept your manuscript and schedule it for publication until you have completed any requested changes.

PRESS

Sincerely, 

Luke

Lucas Smith, Ph.D.

Senior Editor

PLOS Biology

lsmith@plos.org